# Stochastic Approximation to Contrastive Learning

## Abstract

Contrastive learning has proven to be a powerful paradigm for self-supervised representation learning, yet traditional methods often rely on arbitrary definitions of positive and negative pairs, requiring large batch sizes to manage the tradeoff between contrastive terms effectively. This approach wastes significant computational resources on negative pairs that contribute minimal learning signals. To address these limitations, we propose a novel method that reformulates contrastive learning as a matrix approximation problem using I-divergence, a non-normalized variant of Kullback-Leibler divergence. Our objective function is decomposable across instance pairs, enabling efficient stochastic approximation algorithms that perform well with fewer negative samples by leveraging neighbor embeddings. Additionally, we generalize the scaling factor beyond standard normalization to adaptively emphasize positive pairs with higher learning potential, reducing computational waste from negative pairs. Experimental results on benchmark datasets such as CIFAR and ImageNet demonstrate that our method outperforms existing contrastive learning approaches, particularly with small batch sizes and as few as one negative pair, highlighting its effectiveness and computational efficiency.

## 1 Introduction

Learning meaningful representations is fundamental to the success of machine learning systems (Bengio et al., 2013; Goodfellow et al., 2016). While conventional supervised approaches have dominated in recent years, driven by the increasing availability of labeled data, they face scalability challenges and are constrained by the need for extensive labeled examples (LeCun et al., 2015; Balestriero et al., 2023). Self-supervised learning (SSL) has emerged as a transformative paradigm, leveraging the inherent structure of unlabeled data to generate pseudo-labels (Liu et al., 2023; Jing & Tian, 2020; de Sa, 1993; Goyal et al., 2019). In domains such as computer vision, SSL has demonstrated the capability to rival or even surpass supervised pretraining in effectiveness (Tomasev et al., 2022; Goyal et al., 2019; He et al., 2020; Caron et al., 2020; Misra & Maaten, 2020).

Despite its promise, most contrastive learning methods, a cornerstone of SSL, require large batch sizes to manage the balance between positive and negative pairs, leading to significant computational inefficiencies. Specifically, substantial resources are spent on negative pairs that contribute minimal learning signals. Recent advancements, such as SogCLR (Yuan et al., 2022), address some of these inefficiencies by employing an exponentially moving average (EMA) within a decoupled contrastive learning objective (DCL; Yeh et al., 2022), omitting the explicit positive-pair term in their implementation. However, SogCLR introduces additional complexity by requiring EMA-updated scalars for each data instance, which can become inconvenient for large datasets.

To overcome these challenges and enable efficient mini-batch training, we reformulate contrastive learning as a matrix approximation problem using a non-normalized Kullback-Leibler divergence with a scaling factor. Our proposed objective function is decomposable across instance pairs, allowing the development of efficient stochastic approximation algorithms that perform effectively with significantly fewer negative samples. Furthermore, we establish a theoretical connection between our approach and SimCLR while extending beyond it by generalizing the scaling factor. This generalization enables dynamic prioritization of positive pairs that provide richer learning signals, thereby reducing computational waste associated with negative pairs and improving overall efficiency.

We conducted experiments on widely recognized vision benchmark datasets, including CIFAR and ImageNet, to assess the effectiveness of our approach. Our method was evaluated against several state-of-the-art contrastive learning techniques, demonstrating consistent superiority. The results show that our approach is not only more computationally efficient—requiring smaller training batches and fewer negative pairs—but also achieves higher accuracy and overall performance compared to competing methods. These findings underscore the potential of our approach as a cost-effective and high-performing solution for vision-related tasks.

## 2 BACKGROUND AND RELATED WORK

### 2.1 CONTRASTIVE METHODS

Contrastive Learning (CL) (Bromley et al., 1993; Hadsell et al., 2006; Chopra et al., 2005; Gutmann & Hyvärinen, 2010; Mikolov et al., 2013; Oord et al., 2018; Chen et al., 2020a; Sohn, 2016) has historically been influential in representation learning and is based on push- and pull mechanisms between representations. There are supervised approaches (Khosla et al., 2020) which finds positive pairs based on class labels and unsupervised SSL approaches (Chen et al., 2020a; He et al., 2020; Caron et al., 2020; Wu et al., 2018; Misra & Maaten, 2020; Dosovitskiy et al., 2014) which generate positive pairs.

The Contrastive loss (Bromley et al., 1993; Hadsell et al., 2006; Chopra et al., 2005) and Triplet loss (Schroff et al., 2015; Weinberger & Saul, 2009) persisted on mining strategies for tricky negative pairs to be effective. The Triplet loss configured triplets of one positive and negative sample and was generalized to $M$ negative samples instead in Sohn (2016). SimCLR (Chen et al., 2020a) applied this contrastive loss in SSL, which is similar to CPC's (Oord et al., 2018) InfoNCE, in a full batch mode setting where the other positive pairs from the batch configured negative pairs. However SimCLR's CL-loss is not decomposable in minibatch-optimization mode because the negative pairs are weighted relatively in the minibatch (Chen et al., 2022; 2020a). An unfortunate effect of this is that it becomes vulnerable in minibatch-optimization, and this degradation of performance is shown in Chen et al. (2020a). Several CL approaches have resorted to memory-banks (He et al., 2020; Chen et al., 2020b; Caron et al., 2020; Misra & Maaten, 2020) or unreasonable batchsizes (Chen et al., 2020a) and there are sampling strategies (Kalantidis et al., 2020; Robinson et al., 2020), but this can be computationally challenging.

Improving the affordability of contrastive representation learning is still an active research area, and recent related work include among others Yeh et al. (2022); Yuan et al. (2022); Qiu et al. (2023); Chen et al. (2022); Sharma et al. (2023); Shah et al. (2021); HaoChen et al. (2021). DeCL (Chen et al., 2022) consider the non-decomposibility of SimCLR's CL-loss and address the gradient issues. A decomposable spectral contrastive loss was proposed by HaoChen et al. (2021). DCL (Yeh et al., 2022) propose to remove the positive pair from the negative forces which show faster convergence and better results over CL baselines with/without memory-banks. SogCLR (Yuan et al., 2022) propose a global contrastive loss by mixing a zero-initialized running average with a local minibatch-estimate of the negative pairs and shows improvement over SimCLR on ImageNet and CLIP (Radford et al., 2021) in the multi-modal setting. SogCLR is improved in Qiu et al. (2023) with individualized temperatures. AUC-CL (Sharma et al., 2023) propose to combine contrastive learning with AUC maximization.

### 2.2 NON-CONTRASTIVE METHODS

The contrastive methods rely on direct comparisons between positive and negative pairs (Jaiswal et al., 2020) to avoid a degenerate solution which is contrary to the non-contrastive methods which can leverage the positive pairs without directly contrasting to negative pairs (Garrido et al., 2023). Distillation methods (Grill et al., 2020; Chen & He, 2021; Caron et al., 2021; Zhou et al., 2022) exclusively use positive pairs and avoid degenerate solution by adaptions of the architecture while covariance based (Zbontar et al., 2021; Bardes et al., 2021; Ermolov et al., 2021) decorrelate embeddings over the embedding-space (Garrido et al., 2023). The masked learning approaches (He et al., 2022; Zhou et al., 2022) are not based on *multi-view invariance* but typically require certain encoder architectures. Despite self-distillation methods have been effective in SSL there is missing theoretical foundation, and in some areas the architecture cannot be used for example multi-modal

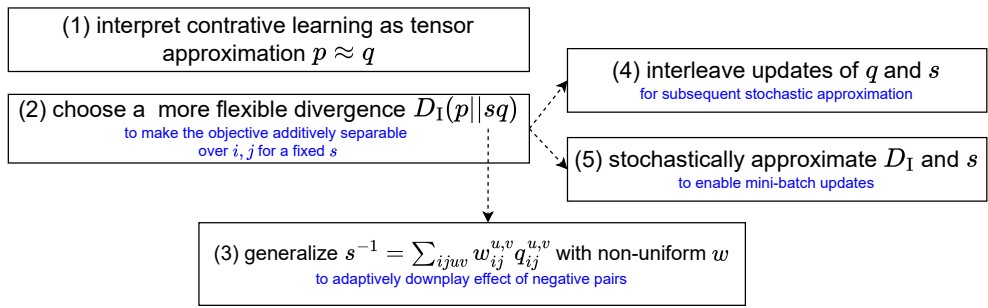

Figure 1: Development clue of our method.

representation learning (Radford et al., 2021; Qiu et al., 2023) and for recent domain agnostic approaches (Sui et al., 2024), where contrastive approaches are more effective.

### 2.3 NEIGHBOR EMBEDDING METHODS

Neighbor embedding (Hinton & Roweis, 2002) (NE) is set of approaches which use the neighborhood graph from the input space as pseudo-labels for the mapped representations. Two pairwise probability matrices $P$ and $Q$ denotes the neighborhood graphs where $P_{ij}$ and $Q_{ij}$ respectively gives the probability of $i$ and $j$ being neighbor in the input-space and representation-space (Hinton & Roweis, 2002; van der Maaten & Hinton, 2008). To obtain this approximation a divergence $D(P||Q)$ is minimized between the pairwise probabilities. Initially SNE (Hinton & Roweis, 2002) normalized the probabilities row-wise which means minimizing a sum of divergences $D(P||Q) = \sum_i D(P_{i:}||Q_{i:})$. In t-SNE (van der Maaten & Hinton, 2008) the probabilities are symmetric, normalized matrix-wise and minimized over one divergence typically Kullback-Leilber divergence $D(P||Q) = \sum_{i \neq j} P_{ij} \log \frac{P_{ij}}{Q_{ij}}$.

It is conceivable a practical difficulty to compute the normalized probabilities $P$ and $Q$. Most solutions can evaluate $P$ offline however the probabilities in $Q$ have to be re-normalized each iteration of optimization. Sampling based approaches UMAP (McInnes et al., 2018), TriMap (Amid & Warmuth, 2019), LargeVis (Tang et al., 2016), SCE (Yang et al., 2023) and PaCMAP (Wang et al., 2021) are not subject to this and can in many cases have better performance than t-SNE (McInnes et al., 2018; Wang et al., 2021; Damrich et al., 2023).

Contrastive learning aspects of NE-method where discussed at three papers at the ICLR 2023 conference by Böhm et al. (2023); Damrich et al. (2023); Hu et al. (2023). Damrich et al. (2023); Hu et al. (2023) discuss how SimCLR's contrastive loss in SSL can be thought of as a parametric t-SNE where the data augmentations are treated as sampling from a nearest graph and show that SimCLR with a Cauchy distribution often used in t-SNE can be very effective for NLDR on complex images where a euclidean metric is inadequate (Bengio et al., 2013). PCA and contrastive learing have been discussed in Tian (2022), and spectral methods and SSL have been brought up in (Balestriero & LeCun, 2022; HaoChen et al., 2021; Tan et al., 2024; Garrido et al., 2023).

## 3 OUR METHOD

### 3.1 REVIEW OF STOCHASTIC CLUSTER EMBEDDING

Our method is inspired by Stochastic Cluster Embedding (SCE; Yang et al., 2023), a similarity-based nonlinear dimensionality reduction (NLDR) method. SCE first computes a similarity matrix $p$ that encodes the pairwise similarities between the high-dimensional data instances $\{\mathbf{x}_i\}_{i=1}^{N}$. Then it finds low-dimensional embedding $\{\mathbf{y}_i\}_{i=1}^{N}$ such that the pairwise similarities in the embedding space $q_{ij} = q(\mathbf{y}_i, \mathbf{y}_j)$ are close to those in the high-dimensional space. Usually $q_{ij} = q(\mathbf{y}_i, \mathbf{y}_j)$ is defined to be $\exp(-\|\mathbf{y}_i - \mathbf{y}_j\|^2)$ or $\frac{1}{1+\|\mathbf{y}_i-\mathbf{y}_j\|^2}$.

SCE employs a more flexible matrix divergence than the Kullback-Leibler (KL) in t-SNE:

$$D_{\text{I}}(p||sq) = \sum_{i=1}^{N} \sum_{j=1, j \neq i}^{N} \left[ p_{ij} \log \frac{p_{ij}}{sq_{ij}} - p_{ij} + sq_{ij} \right], \tag{1}$$

where $s > 0$ is a scaling factor. The divergence is called non-normalized KL-divergence or I-divergence. It measures the discrepancy between two matrices up to a scale. When $s^{-1} = \sum_{ij} q_{ij}$, minimizing the I-divergence reduces to minimizing the normalized KL-divergence. Differently, SCE periodically updates $s^{-1} = \sum_{ij, i \neq j} w_{ij} q_{ij}$, where $w_{ij} = \alpha p_{ij} N(N-1) + (1-\alpha)$ with $\alpha \in [0, 1]$. When $\alpha = 0$, it reduces to the t-SNE choice. When $\alpha > 0$, it mixes $p$ and uniform sampling in calculating $s$, which adaptively reduces repulsion and thus improves cluster visualization (Yang et al., 2023).

## 3.2 Contrastive Learning Loss Function

SCE cannot be directly applied to contrastive learning due to several limitations: (1) SCE relies on fixed input similarities and cannot generalize to data points outside the training set; (2) its similarities are typically computed using simple metrics like Euclidean distances, which are inadequate for complex data objects such as images; and (3) SCE lacks mini-batch training algorithms tailored for contrastive learning. To address these issues, we introduce three key modifications: (1) we parameterize the embedding function using a deep neural network, $\mathbf{y} = f(\mathbf{x}; \theta)$, where $\theta$ represents the network weights; (2) we define similarities based on data and augmented view indices, eliminating the dependency on Euclidean or similar metrics for neighbor search before embedding; and (3) we propose stochastic approximation algorithms for the resulting learning objective and scaling factor. The progression of our development is illustrated in Figure 1.

For each input $\mathbf{x}_i$, we obtain its two augmented views $\tilde{\mathbf{x}}_i^{(1)}$ and $\tilde{\mathbf{x}}_i^{(2)}$ (one of the augmentation can be identity transform). Denote $\tilde{\mathbf{y}}_i^{(u)} = f\left(\tilde{\mathbf{x}}_i^{(u)}; \theta\right)$ and $q_{ij}^{u,v} = q\left(\tilde{\mathbf{y}}_i^{(u)}, \tilde{\mathbf{y}}_j^{(v)}\right)$ for $i, j \in \{1, \dots, N\}$ and $u, v \in \{1, 2\}$. In contrastive learning, a well-trained neural network should make the outputs $\tilde{\mathbf{y}}_i^{(1)}$ and $\tilde{\mathbf{y}}_i^{(2)}$ from the same input instance $\mathbf{x}_i$ to be similar and the outputs from different instances (i.e., $i \neq j$) to be dissimilar. If we specify the similar target to be 1 and dissimilar to be 0, we can write the targets in a four-dimensional tensor $p \in \mathbb{R}^{N \times N \times 2 \times 2}$, where $p_{ii}^{1,2} = p_{ii}^{2,1} = 1$ for $i = 1, \dots, N$ and otherwise 0. Alternatively, we can reorgainze the tensors $p$ and $q$ in two $2N \times 2N$ matrices where $\psi_{2(i-1)+u, 2(j-1)+v} = p_{ij}^{u,v}$ and $\phi_{2(i-1)+u, 2(j-1)+v} = q_{ij}^{u,v}$. Then we can formulate contrastive learning as a matrix approximation problem by minimizing $D_{\text{I}}(\psi||s\phi)$.

We neglect the matrix diagonal of $\psi$ and $\phi$ in the approximation because $q_{ii}^{u,u}$'s are always a constant for Gaussian or Cauchy kernels. For notational simplicity, we set $p_{ii}^{u,u} = q_{ii}^{u,u} = 0$ for $i = 1, \dots, N$ and $u = 1, 2$, i.e., $\psi_{aa} = \phi_{aa} = 0$ for $a = 1, \dots, 2N$, which is equivalent to excluding them from the summations.

Because there are only a few nonzeros in $\psi$, we can rewrite $D_{\text{I}}(\psi||s\phi)$ as

$$\mathcal{L}_{\text{CLR}}(\theta) = \sum_{a=1}^{2N} \sum_{b=1}^{2N} \left[ \psi_{ab} \log \frac{\psi_{ab}}{s\phi_{ab}} - \psi_{ab} + s\phi_{ab} \right] \tag{2}$$

$$= \sum_{i=1}^{N} -2 \log q_{ii}^{1,2} + s \sum_{i=1}^{N} \sum_{j=1}^{N} \sum_{u=1}^{2} \sum_{v=1}^{2} q_{ij}^{u,v} + C_1, \tag{3}$$

where $C_1 = -2N - 2N \log s$ is a constant for a fixed $s$. Following SCE, we periodically updates $s^{-1} = \sum_{ijuv} w_{ij}^{u,v} q_{ij}^{u,v}$ with

$$w_{ij}^{u,v} = \alpha p_{ij}^{u,v} N + (1-\alpha) = \begin{cases} \alpha N + (1-\alpha) & \text{when } i = j \\ 1 - \alpha & \text{otherwise.} \end{cases} \tag{4}$$

When $\alpha = 0$, $s^{-1}$ becomes $\sum_{ijuv} q_{ij}^{u,v}$ and leads to normalized KL-divergence between $\psi$ and $\phi$.

This generalization brought by $w$ can adaptively change the tradeoff between the first two terms in $\mathcal{L}_{\text{CLR}}(\theta)$. At the training start, the non-zero $q$ entries do not differ much, and thus $s^{-1} \approx \sum_{ijuv} q_{ij}^{u,v}$.

After minimizing the discrepancy for a while, $q_{ii}^{u,v}$'s ($u \neq v$) will become larger because of approximating $p$. When $\alpha > 0$, the matrix $w$ concentrates more on the entries where $i = j$, and the resulting $s^{-1}$ becomes larger than the uniform choice of $w$ (i.e., $\alpha = 0$ or the normalized KL). Therefore, the generalization with $\alpha > 0$ dynamically emphasizes the first term, which contains the learning signals $p$, and downplays the second term without learning signals.

### 3.3 MINIBATCH-MODE OPTIMIZATION

It is expensive to directly optimize the CL objective because it requires all pairs of data instances. Stochastic approximation is needed to facilitate minibatch-mode optimization. We first rewrite $\mathcal{L}_{\text{CLR}}(\theta)$ in an expectation manner

$$\mathcal{L}_{\text{CLR}}(\theta) = N \mathbb{E}_{i \sim \text{Uniform}(\{1,\ldots,N\})} \left\{ -2 \log q_{ii}^{1,2} + sN \mathbb{E}_{j \sim \text{Uniform}(\{1,\ldots,N\})} \left\{ \sum_{u=1}^{2} \sum_{v=1}^{2} q_{ij}^{u,v} \right\} \right\}. \quad (5)$$

Our method, named Stochastic Approximation to Contrastive Learning (SACLR), minimizes the following minibatch-mode objective

$$\mathcal{L}_{\text{SACLR}}(\theta) = \sum_{i \in \mathcal{B}} \left[ -2 \log q_{ii}^{1,2} + s \frac{N}{M} \sum_{j \in \mathcal{M}_i} \sum_{u=1}^{2} \sum_{v=1}^{2} q_{ij}^{u,v} \right], \quad (6)$$

where $\mathcal{B} = \{i_1, \ldots, i_B\} \subset [1, \ldots, N]$, $\mathcal{M}_i \subseteq \mathcal{B}$, and $M = |\mathcal{M}_i|$ for all $i$. We have studied two choices of $\mathcal{M}_i$'s: (1) SACLR-1 where $M = 1$, and the objective simplifies to $\sum_{i \in \mathcal{B}} \left[ -2 \log q_{ii}^{1,2} + sN \sum_{uv} q_{ij}^{u,v} \right]$ with $j \sim \mathcal{B}$; and (2) SACLR-all where $M = B$ uses all negative pairs in the batch. The gradients of $\mathcal{L}_{\text{SACLR}}(\theta)$ can then be used in Stochastic Gradient Descent or Adam-style optimization (Kingma & Ba, 2014).

### 3.4 EXPONENTIAL MOVING AVERAGE OF $s$

The scaling factor $s$ can also be calculated in an expectation manner:

$$s^{-1} = \sum_{i=1}^{N} \sum_{j=1}^{N} \sum_{u=1}^{2} \sum_{v=1}^{2} \left[ \alpha p_{ij}^{u,v} N + (1-\alpha) \right] q_{ij}^{u,v} \quad (7)$$

$$= N^2 \mathbb{E}_{i \sim \text{Uniform}(\{1,\ldots,N\})} \left\{ 2\alpha q_{ii}^{1,2} + (1-\alpha) \mathbb{E}_{j \sim \text{Uniform}(\{1,\ldots,N\})} \left\{ \sum_{u=1}^{2} \sum_{v=1}^{2} q_{ij}^{u,v} \right\} \right\}. \quad (8)$$

Therefore $\xi = \frac{N^2}{B} \sum_{i \in \mathcal{B}} \left( 2\alpha q_{ii}^{1,2} + (1-\alpha) \frac{1}{M} \sum_{j \in \mathcal{M}_i} \sum_{u=1}^{2} \sum_{v=1}^{2} q_{ij}^{u,v} \right)$ is the stochastic approximation of $s^{-1}$ in a minibatch. We can then use an exponential moving average to update the estimate of $s^{-1} \leftarrow \rho s^{-1} + (1-\rho)\xi$ with a forgetting rate $\rho \in (0,1)$ after each batch.

### 3.5 APPROXIMATION TO MATRIX ROWS

We have derived SACLR with matrix-wise approximation to $\psi$ by following SCE in the above. Next, we present the row-wise approximation to $\psi$, which gives a more direct connection to the cross-entropy based contrastive InfoNCE loss functions practised by the existing SimCLR and DCL (Yeh et al., 2022) unbounded by the number of negative samples.

We define $\forall a$, $\psi_{a:} = \{\psi_{ab} | b \in \{1, \ldots, N\} \setminus \{a\}\}$, i.e., the $a$-th row of $\psi$ with the diagonal element excluded. $\phi_{a:}$ is similarly defined over $\phi$. We can then apply $D_{\text{I}}(\psi_{a:} || s_a \phi_{a:})$ to measure the discrepancy between $\psi_{a:}$ and $\phi_{a:}$ up to a scaling $s_a$. The contrastive learning function becomes

$$\mathcal{L}_{\text{CLR-row}}(\theta) = \sum_{a=1}^{2N} D_{\text{I}}(\psi_{a:} || s_a \phi_{a:}) \quad (9)$$

$$= \sum_{i=1}^{N} \left[ -2 \log q_{ii}^{1,2} + \sum_{u=1}^{2} s_{2(i-1)+u} \sum_{j=1}^{N} \sum_{v=1}^{2} \mathbb{1}_{[i \neq j \text{ or } u \neq v]} q_{ij}^{u,v} \right] + C_2 \quad (10)$$

---

**Algorithm 1** SACLR algorithm (using matrix row approximation)

---

**Input:** Input data $\{\mathbf{x}_i\}_{i=1}^N$, weighting rate $\alpha \in [0, 1]$, forgetting rate $\rho \in (0, 1)$, number of iterations $T$, batch size $B$, number of negative samples $M$, and a neural network $f$ (parameterized by $\theta$).

1: Initialize the neural network $f$, and $s_i \leftarrow 1/N$ for $i = 1, \ldots, N$
2: **for** $t = 1, \ldots, T$ **do**
3:     Uniformly draw $\mathcal{B} = \{i_1, \ldots, i_B\}$ from $\{1, \ldots, N\}$
4:     Augment $\mathbf{x}_i$ to $\tilde{\mathbf{x}}_i^{(1)}$ and $\tilde{\mathbf{x}}_i^{(2)}$ for $i \in \mathcal{B}$
5:     $\mathcal{L} \leftarrow 0$
6:     **for** $i \in \mathcal{B}$ **do**
7:         Compute $q_{ii}^{1,2} = q\big(f(\tilde{\mathbf{x}}_i^{(1)}; \theta), f(\tilde{\mathbf{x}}_i^{(2)}; \theta)\big)$
8:         Uniformly draw $\mathcal{M}_i = \{j_1, \ldots, j_M\}$ from $\mathcal{B}$
9:         Compute $q_{ij}^{u,v} = q\big(f(\tilde{\mathbf{x}}_i^{(u)}; \theta), f(\tilde{\mathbf{x}}_j^{(v)}; \theta)\big)$ for $j \in \mathcal{M}_i$, $u \in \{1, 2\}$, and $v \in \{1, 2\}$
10:        $\mathcal{L} \leftarrow -2 \log q_{ii}^{1,2}$
11:        **for** $u \in \{1, 2\}$ **do**
12:            $\mathcal{L} \leftarrow \mathcal{L} + s_{2(i-1)+u} \frac{N}{M} \sum_{j \in \mathcal{M}_i} \sum_{u=1}^2 \sum_{v=1}^2 q_{ij}$
13:            $\xi_{2(i-1)+u} \leftarrow N(2\alpha q_{ii}^{1,2} + (1-\alpha) \frac{1}{M} \sum_{j \in \mathcal{M}_i} \sum_{v=1}^2 q_{ij}^{u,v})$
14:            $s_{2(i-1)+u}^{-1} \leftarrow \rho s_{2(i-1)+u}^{-1} + (1-\rho) \xi_{2(i-1)+u}$
15:        **end for**
16:    **end for**
17:    Update $\theta$ with $\nabla_\theta \mathcal{L}$ using an SGD or Adam-style step.
18: **end for**
**Output:** Trained neural network $f$.

---

where $\mathbb{1}_{[\cdot]} = 1$ when the bracketed condition is true and otherwise 0, and $C_2 = -2N - \sum_{a=1}^{2N} \log s_a$ is a constant for fixed $s_a$'s.

The row approximation objective is connected to SimCLR (proof in Appendix A.5):

**Theorem 1.** *When $s_{2(i-1)+u}^{-1} = \sum_{jv} \mathbb{1}_{[i \neq j \text{ or } u \neq v]} q_{ij}^{u,v}, \forall i, u$, minimizing $\mathcal{L}_{CLR\text{-}row}(\theta)$ is equivalent to minimizing[1] $\mathcal{L}_{SimCLR}(\theta) = -\sum_{i=1}^N \sum_{u=1}^2 \log \frac{q_{ii}^{1,2}}{\sum_{j=1}^N \sum_{v=1}^2 \mathbb{1}_{[i \neq j \text{ or } u \neq v]} q_{ij}^{u,v}}$.*

Differently, we employ a generalization $s_{2(i-1)+u}^{-1} = \sum_{jv} = w_{ij}^{u,v} q_{ij}^{u,v}$ with non-uniform $w$ defined in Eq. 4. The change adaptively reduces the effect of negative pairs and leads to improvements in our experiments.

The row-wise approximation version of SACLR objective is

$$\mathcal{L}_{\text{SACLR-row}}(\theta) = \sum_{i \in \mathcal{B}} \left[ -2 \log q_{ii}^{1,2} + \sum_{u=1}^2 s_{2(i-1)+u} \frac{N}{M} \sum_{j \in \mathcal{M}_i} \sum_{v=1}^2 q_{ij}^{u,v} \right] \tag{11}$$

The expectation form of $s_{2(i-1)+u}$ equals $N\left[\alpha q_{ii}^{1,2} + (1-\alpha)\mathbb{E}_{j \sim \text{Uniform}(\{1,\ldots,N\})} \left\{ \sum_{v=1}^2 q_{ij}^{u,v} \right\} \right]$, and its stochastic approximation is $\xi_{2(i-1)+u} = N\left(\alpha q_{ii}^{1,2} + (1-\alpha)\frac{1}{M} \sum_{j \in \mathcal{M}_i} \sum_{v=1}^2 q_{ij}^{u,v}\right)$. with the EMA update rule $s_{2(i-1)+u}^{-1} \leftarrow \rho s_{2(i-1)+u}^{-1} + (1-\rho)\xi_{2(i-1)+u}$.

This version of SACLR require $2N$ scaling factors. In practice, the two scaling factors of each data instance will be very similar. Therefore we can ease the requirement by only using $N$ scaling factors in the actual implementation. The same strategy is also used by SogCLR (Yuan et al., 2022) and iSogCLR's (Qiu et al., 2023). The pseudocode of SACLR using matrix row approximation is given in Algorithm 1. The version using matrix-wise approximation is similar and given in Appendix A.6.

---

[1]The original SimCLR use exponential over cosine similarities. They are equivalent to the Gaussian kernels given that $\tilde{\mathbf{y}}_i^{(1)}$'s and $\tilde{\mathbf{y}}_i^{(2)}$'s are normalized.

Table 1: Top-1 linear validation accuracy on ImageNet100. * marks improved results reported by Qiu et al. (2023). Standard deviations are from three different runs.

| Method | Batchsize | 400EP |
|---|---|---|
| SACLR-1 matrix (ours) | 128 | **82.30** ($\pm$ 0.46) |
| SACLR-1 row (ours) | 128 | 81.59 ($\pm$ 0.07) |
| SimCLR (Chen et al., 2020a) * | 256 | 79.96 ($\pm$ 0.2) |
| SogCLR (Yuan et al., 2022) * | 256 | 80.54 ($\pm$ 0.14) |
| iSogCLR (Qiu et al., 2023) * | 256 | 81.14 ($\pm$ 0.19) |

Table 2: Top-1 linear validation accuracy on ImageNet1k. Standard deviations are from three different runs.

| Method | Batchsize | 100EP | 400EP |
|---|---|---|---|
| SACLR-1 matrix (ours) | 128 | **64.94** $\pm$ (0.16) | **67.57** ($\pm$ 0.01) |
| SACLR-1 row (ours) | 128 | 64.78 ($\pm$ 0.05) | 67.50 ($\pm$ 0.14) |
| SimCLR (Chen et al., 2020a) | 256 | 62.80 | 65.70 |
| SogCLR (Yuan et al., 2022) | 128 | 64.90 | 67.40 |

## 4 EXPERIMENTS

We employ our methods on color images from ImageNet (Russakovsky et al., 2015) and CIFAR (Alex, 2009). The image augmentations and network architecture on ImageNet follows SimCLR (Chen et al., 2020a), LARS-optimizer (You et al., 2017) and the projector from VIC-REG (Bardes et al., 2021). Evaluation follows the linear evaluation protocol on frozen representations from the backbone. Full implementation details on ImageNet are described in Appendix A.1 and on CIFAR in Appendix A.2. While the introduced algorithms can work with $M > 1$ negative samples we focus on SACLR with $M = 1$ and use $M \gg 1$ for ablations.

**ImageNet:** We find that our methods achieve good results from image SSL-pretraining on a merited image-dataset as ImageNet. The results show that our methods can consistently have improvements to many existing CL-approaches in the low batchsize setting and in addition can be more memory efficient. On ImageNet we compare our methods to existing CL approaches such as SimCLR (Chen et al., 2020a), and recent stochastic estimation based CL approaches SogCLR (Yuan et al., 2022) and iSogCLR (Qiu et al., 2023). All methods are pretrained for different amounts of epochs and each reported value is from one run from scratch. The linear classification accuracies after image-pretraining on ImageNet100 (Wu et al., 2019) are presented in Table 1 and the results on ImageNet1k (Russakovsky et al., 2015) are presented in Table 2. We exclusively report values from each methods respective paper unless explicitly mentioned. The full results from image pretraining on ImageNet1k and ImageNet100 are presented in Table 5 and Table 6 in Appendix. Remarkably we see that with $M = 1$ negative samples per anchor SACLR can give performance improvements to SimCLR and SogCLR-variants which use $M \gg 1$ on both ImageNet1k and ImageNet100. We prove that our methods are also very effective for unsupervised cluster visualization in Figure 2.

**CIFAR:** We also include results with a ResNet18 encoder on CIFAR and Imagenette (10 class subset of ImageNet; Howard, 2019) in Tables 3 and 4. We pretrain for 1000 epochs on CIFAR and 800 epochs on Imagenette. On CIFAR the classifer is a linear layer and we include reported values of SimCLR and SimSiam from solo-learn da Costa et al. (2022). On Imagenette we use 20NN classifier from scikit-learn (Pedregosa et al., 2011) with cosine-weighting $w = \exp(\text{CosSim}(\mathbf{a}, \mathbf{b})/0.07)$ (Wu et al., 2018; Caron et al., 2021; Balestriero et al., 2023) and find reported values from Lightly AI [2].

Whereas existing approaches in CL typically are advised to find $M \gg 1$ negative samples per anchor (Chen et al., 2020a; He et al., 2020; Yeh et al., 2022; Damrich et al., 2023) or require additional variables to network parameters which scale with the number of input-instances $N$ (Yuan et al., 2022; Qiu et al., 2023), our methods can leverage $M = 1$ negative sample per anchor and a single additional variable devoted for the stochastic estimation. A more comprehensive overview of these

---

[2]`https://docs.lightly.ai/self-supervised-learning/getting_started/benchmarks.html`

Table 3: Top 1 accuracy with a linear classifier on CIFAR. Results marked with $^*$ are from solo-learn (da Costa et al., 2022) and $^{**}$ are from Qiu et al. (2023). Standard deviations are from three different runs.

| Method | CIFAR10 | CIFAR100 |
|---|---|---|
| SACLR-1 matrix (ours) | **93.07** ($\pm$ 0.17) | 70.22 ($\pm$ 0.38) |
| SACLR-1 row (ours) | 92.86 ($\pm$ 0.08) | 70.39 ($\pm$ 0.3) |
| SACLR-all matrix (ours) | 92.91 | 70.89 |
| SACLR-all row (ours) | 92.98 | **71.50** |
| SimCLR (Chen et al., 2020a) $^*$ | 90.74 | 65.78 |
| SogCLR (Yuan et al., 2022)$^{**}$ | 90.07 ($\pm$ 0.10) | 65.18 ($\pm$ 0.10) |
| iSogCLR (Qiu et al., 2023)$^{**}$ | 90.25 ($\pm$ 0.09) | 65.95 ($\pm$ 0.07) |
| SimSiam (Chen & He, 2021) $^*$ | 90.51 | 66.04 |
| ReSSL (Zheng et al., 2021) $^*$ | 90.63 | 65.92 |

Table 4: Top-1 accuracy using a 20NN-classifier for CIFAR and Imagenette datasets.

| Method | CIFAR10 | CIFAR100 | Imagenette |
|---|---|---|---|
| SACLR-1 matrix (ours) | 91.65 ($\pm$ 0.01) | 67.02 ($\pm$ 0.21) | **90.74** ($\pm$ 0.04) |
| SACLR-1 row (ours) | 91.75 ($\pm$ 0.06) | 67.33 ($\pm$ 0.03) | 90.68 ($\pm$ 0.2) |
| SACLR-all matrix (ours) | 91.83 | 67.89 | - |
| SACLR-all row (ours) | **91.87** | 68.05 | - |
| SimCLR (Chen et al., 2020a) | 90.59 $^*$ | 65.32 $^*$ | 88.90 |
| SogCLR (Yuan et al., 2022) | 89.98 | 65.95 | - |
| iSogCLR (Qiu et al., 2023) | 91.69 | **68.85** | - |
| SimSiam (Chen & He, 2021) | 90.83 $^*$ | 66.43 $^*$ | 87.20 |
| ReSSL (Zheng et al., 2021) | 90.80 $^*$ | 66.08 $^*$ | - |

memory requirements are demonstrated in Table 7 in Appendix. More tables of runtime and memory usage in Appendix A.3 are found in Table 8, 9 and 10.

## 5 ABLATION STUDIES

We perform ablations to see the effect of different forgetting rates, weighting rates, global compared to individualized adaptive scaling parameters and $M$ negative samples per anchor in Appendix A.4. The findings are that our methods are robust against hyperparameters and fewer negative samples.

This work presented a matrix-method with a one global adaptive scaling factor and a row-method with individualized adaptive scaling factors. We study the impact of the respective methods in Table 14 in Appendix. Surprisingly we find that the matrix-method performs better or evenly to the row-method which has a cost of more variables. Visual example is shown in Figure 3 where we log the positive and negative values with the estimated partition function(s) after each epoch. While the row-method can offer a more specific estimate especially in the early stages it might seem to converge to a similar value in the later stages. We also observe a negligible difference between SACLR with $M = 1$ negative sample compared to the full batch mode version where $M = 2 \times 128 - 2$.

The hyperparameters of most importance are the forgetting rate $\rho$ and weighting rate $\alpha$, in addition to initialization of scaling factors. We tune the forgetting rate $\rho$ on Imagenette in Table 12 and Table 13 in Appendix. We find that $\rho = 0.99$ works best for the matrix-method and $\rho = 0.9$ for the row-method. The matrix-method re-estimates the scaling factor after each minibatch which is contrary to the row-method which re-estimates per instance approximately at each epoch, see Alg.(1) and Alg.(2) for more details. A consequence of this can be a necessity to use a higher forgetting rate, which is demonstrated in Table 13 with a linear classifier. The weighting rate $\alpha$ is ablated in Table 15 on ImageNet for SACLR-1. We initially set $\alpha = 0.5$ and see that setting a lower weighting-rate $\alpha = 0.125$ gives better performance.

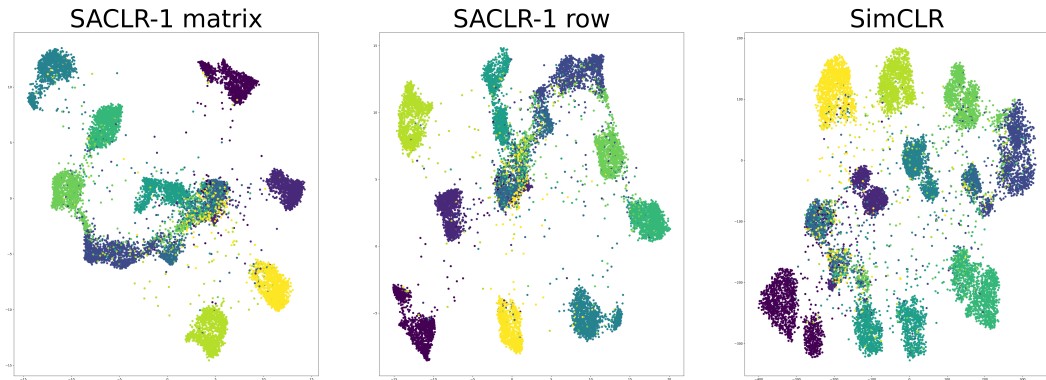

Figure 2: Embedding visualizations of Imagenette dataset. All compared methods used the Cauchy kernel and were trained with raw images. We followed the strategies by Böhm et al. (2023): we first pretrained the models in 8192-dimensional space over 800 epochs and then finetuned a 2D projector over 250 epochs. SACLR-1 used $M = 1$ negative sample while SimCLR used $M = 128 \times 2 - 2$ for each image in a batch.

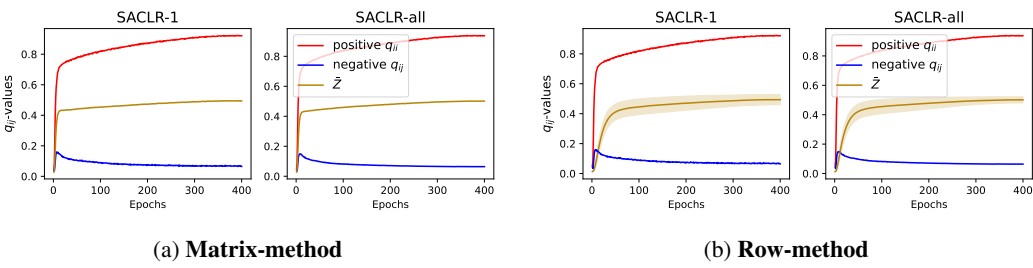

Figure 3: Optimization dynamics: positive and negative values with estimated partition function(s) on Imagenette. We include the variability for the row method with standard deviations.

## 6 CONCLUSION AND FUTURE WORK

We have approached contrastive learning as a matrix approximation problem. A key component of our method is an adaptive scaling factor, which optimizes separable similarities and enables memory-efficient stochastic approximation algorithms. This allows our approach to require significantly smaller training batches and as few as one negative pair, making it substantially more efficient than other contrastive learning techniques. Despite its economical design, our method has demonstrated the effectiveness of SACLR on CIFAR and ImageNet classification tasks, with consistent performance improvements to the compared methods.

In the future, our work could be extended in several promising directions. One potential area of improvement is integrating contextual learning within individual data instances, in addition to the contrastive learning across data populations. By capturing more detailed relationships within a single data sample, such as spatial or sequential patterns, we could significantly enhance the model's ability to learn richer, more nuanced representations. This integration could improve performance in tasks that require fine-grained understanding, like object detection or temporal data analysis.

Another direction is replacing the traditional convolutional neural network with more advanced architectures, such as self-attention-based models like the Transformer. Self-attention mechanisms have shown remarkable success in capturing long-range dependencies and global patterns in various domains. Incorporating this into our framework could lead to more expressive models that can handle more complex data types, such as video or multi-modal inputs, while further improving efficiency and performance.

REPRODUCIBILITY STATEMENT

The code is made available and the README includes examples of how to reproduce the results. In addition we provide description of data processing and implementation details in A.1 and A.2.

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

# A  APPENDIX

## A.1  IMPLEMENTATION DETAILS IMAGENET

### A.1.1  DATASETS AND AUGMENTATIONS

ImageNet is an eminent image-dataset and contains color images from rich range of different classes. This work use the subsets ImageNet1k (Russakovsky et al., 2015) (ILSVRC2012) which contains 1,281,167 training images from 1000 different classes, and ImageNet100 (Wu et al., 2019) which contains 126,689 training images from 100 different classes. The pre-defined training and validation split are used, and we evaluate our methods on the validation set, similar to existing work. For ablations and hyperparameter tuning shorter runs and the image-dataset Imagenette (Howard, 2019) are used. Imagenette is a subset from ImageNet with 10 different classes. All images from ImageNet are resized to two $(224 \times 224)$ image views without any multi-crop from Caron et al. (2020). All images are applied the augmentations from SimCLR (Chen et al., 2020a) during pretraining, and we use the augmentation implementations from torchvision. We download ImageNet (ILSVRC2012) from the official website and apart from that use the built-in datasets to torchvision.

### A.1.2  ARCHITECTURE

The neural network architecture in this work adhere to SimCLR (Chen et al., 2020a), and contains a backbone encoder and a projector. The backbone encoder is a ResNet50 (He et al., 2016) and the projector is the 3-layered MLP from VIC-REG (Bardes et al., 2021). The projector is applied on the output of the ResNet's avg-pooling layer and the output of each linear layer is 8192-d. We use the ResNet architecture from torchvision.

### A.1.3  OPTIMIZATION

The neural network optimizer in this work is the LARS-optimizer (You et al., 2017) with the square-root learning rate scaling for low batchsizes proposed by SimCLR (Chen et al., 2020a) $lr = \sqrt{batchsize} \times 0.075$, and weight decay $wd = 10^{-4}$. The learning rate is linearly warmed up the 10 first epochs and then annealed with a cosine schedule (Loshchilov & Hutter, 2016) until it reaches $lr/1000$. The default batchsize is 128.

### A.1.4  LINEAR EVALUATION

Evaluation follows the linear classifier protocol on frozen backbone representations (Zhang et al., 2016; Balestriero et al., 2023). This work keep the optimization settings from Caron et al. (2020); Zbontar et al. (2021) with SGD-optimizer, momentum $= 0.9$, $wd = 10^{-6}$ and $lr = 0.3$. The learning rate is annealed with a cosine-scheduler down to zero over 100 epochs. During training we use random-cropping, -resizing and horizontal flipping, and we test with a center crop.

### A.1.5  LOSS FUNCTION HYPERPARAMETERS

This work set by default weighting rate $\alpha = 0.125$. The forgetting rate is tuned $\rho \in \{0.9, 0.99, 0.999\}$ on Imagenette. The adaptive scaling parameter(s) are initialized such that the partition function(s) at start correspond to 0.01, e.g. $s = N^{-2}10^2$ where $N$ is the number of training images.

### A.1.6  SIMILARITY FUNCTION

The similarity function in this work is a squared exponential kernel

$$q(\mathbf{y}_i, \mathbf{y}_j) = \exp(-\|\overline{\mathbf{y}}_i - \overline{\mathbf{y}}_j\|^2/2\tau^2) \tag{12}$$

on the unit-sphere normalized neural network output $\overline{\mathbf{y}_i} = \mathbf{y}_i/\|\mathbf{y}_i\|$ and $\overline{\mathbf{y}_j} = \mathbf{y}_j/\|\mathbf{y}_j\|$, with a chosen temperature $\tau \in \mathbb{R}_+$. In this work we set $\tau = 0.5$.

Table 5: **ImageNet100:** Top 1 linear validation accuracy on ImageNet100 with ResNet50. $*$ is a mark of improved results reported by Qiu et al. (2023). Standard deviations are from three different runs.

| Method | Batchsize | 400EP |
|---|---|---|
| SACLR-1 matrix (ours) | 128 | 82.30 ($\pm$ 0.46) |
| SACLR-1 row (ours) | 128 | 81.59 ($\pm$ 0.07) |
| SACLR-all matrix (ours) | 128 | 82.50 |
| SACLR-all row (ours) | 128 | 82.18 |
| SimCLR (Chen et al., 2020a) $*$ | 256 | 79.96 ($\pm$ 0.2) |
| SogCLR (Yuan et al., 2022) $*$ | 256 | 80.54 ($\pm$ 0.14) |
| iSogCLR (Qiu et al., 2023) $*$ | 256 | 81.14 ($\pm$ 0.19) |

Table 6: **ImageNet1k:** Top 1 linear validation accuracy on ImageNet1k with ResNet50. Standard deviations are from three different runs.

| Method | Batchsize | 100EP | 400EP |
|---|---|---|---|
| SACLR-1 matrix (ours) | 128 | 64.94 ($\pm$ 0.16) | 67.57 ($\pm$ 0.01) |
| SACLR-1 row (ours) | 128 | 64.78 ($\pm$ 0.05) | 67.50 ($\pm$ 0.14) |
| SACLR-all row (ours) | 128 | 65.90 | 67.34 |
| SACLR-1 row (ours) | 256 | 65.35 | - |
| SACLR-all row (ours) | 256 | 66.75 | - |
| SimCLR (Chen et al., 2020a) | 256 | 62.80 | 65.70 |
| SogCLR (Yuan et al., 2022) | 128 | 64.90 | 67.40 |

### A.1.7 COMPUTE REQUIREMENTS

This work use exclusively one GPU setups with at least 24 GB VRAM. On ImageNet we pretrain with Nvidia H100 and Nvidia A100 GPUs on a SLURM cluster, and use a Nvidia RTX4090 for other datasets.

### A.2 IMPLEMENTATION DETAILS CIFAR

CIFAR (Alex, 2009) contains downsampled ($32 \times 32$) color images with 10 different classes in CI-FAR10 and 100 different classes in CIFAR100. The experimental setup on CIFAR is more or less the same to the setup on ImageNet in A.1 but add a few alternations to downsampled ($32 \times 32$) images following Chen et al. (2020a); Chen & He (2021). The ResNet-architecture and image augmentations are adapted to CIFAR (Chen et al., 2020a). The optimization settings for unsupervised pretraining and linear frozen evaluation are the same to Chen & He (2021) on CIFAR10.

### A.3 ADDITIONAL RESULTS

We compare our methods to existing CL methods as SimCLR (Chen et al., 2020a) and its recent improvements SogCLR (Yuan et al., 2022) and iSogCLR (Qiu et al., 2023) for low batchsize settings. All methods use ResNet50 backbone and augmentation strategies from Chen et al. (2020a) on two ($224 \times 224$) image views. The full results from the linear classifier after image-pretraining on ImageNet100 are shown in Table 5 and on ImageNet1k in Table 6. Results of SACLR use $\alpha = 0.125$. We provide an overview of computational costs and additional memory requirements for our own methods and existing stochastic estimation based CL-approaches in Table 7, 8, 9 and 10.

### A.4 ABLATION RESULTS

We continue from Section 5 with more details of investigations into effects of number of negative samples, approximation-methods and choice of hyperparameters. By default we set $\alpha = 0.5$ for these trials. The forgetting rate is tuned $\rho \in \{0.9, 0.99, 0.999\}$ on Imagenette (Howard, 2019), where we pretrain for 400 epochs and use a separate validation set split (20%) from the training set.

Table 7: **Memory complexity and additional memory complexity to network parameters**. We compare memory efficiency and required extra variables for the stochastic estimation between CL methods. We denote the number of input instances by $N$, negative samples per anchor $M$ and batchsize $B$.

| Method | $M$ | Additional variables |
|---|---|---|
| SACLR-1 matrix-method (ours) | 1 | 1 |
| SACLR-1 row-method (ours) | 1 | $N$ |
| SACLR-all matrix-method (ours) | $2B - 2$ | 1 |
| SACLR-all row-method (ours) | $2B - 2$ | $N$ |
| SimCLR (Chen et al., 2020a) | $2B - 2$ | 0 |
| SogCLR (Yuan et al., 2022) | $2B - 2$ | $N$ |
| iSogCLR (Qiu et al., 2023) | $2B - 2$ | $4N$ |

Table 8: **Computational costs on CIFAR**. GPU peak memory usage over large batch training over ranges $B$ often used with the LARS (You et al., 2017) optimizer. In this study the network is a ResNet18 on CIFAR10 over 10EP. The last fc-layer has 128 units. We downloaded the PyTorch implementations from SogCLR (Yuan et al., 2022) and iSogCLR (Qiu et al., 2023) in this study. Experiments conducted on a 10 core CPU with 32GB RAM and NVIDIA H100 80GB VRAM GPU. 1K denote 1024 and values over 80GB are OOM-errors.

| Method $\backslash B$ | 128 | 1K | 2K | 4K | 8K | 16K | 32K |
|---|---|---|---|---|---|---|---|
| SACLR-1 | 1.17 GB | 2.69 GB | 4.38 GB | 7.59 GB | 14.35 GB | 27.12 GB | 53.17 GB |
| SACLR-all | 1.17 GB | 2.73 GB | 4.58 GB | 8.50 GB | 17.73 GB | 39.44 GB | $> 80$GB |
| SimCLR | 1.17 GB | 2.69 GB | 4.43 GB | 7.94 GB | 15.48 GB | 30.42 GB | $> 80$GB |
| SogCLR | 1.17 GB | 2.78 GB | 4.62 GB | 8.74 GB | 17.70 GB | 50.92 GB | $> 80$GB |
| iSogCLR | 1.17 GB | 2.73 GB | 4.50 GB | 7.99 GB | 14.72 GB | 38.92 GB | $> 80$GB |

We report accuracy with the linear classifier in Table 12 and Table 13. The impact of applying row-based estimation compared to matrix-based estimation is studied in ImageNet1k over 100 epoch runs and ImageNet100 with a linear classifier in Table 14. Notably we see that the number of negative samples do not leave a significant impact on SACLR from Table 12, Table 13 and Table 14. The weighting term is ablated with a linear classifier over shorter 100 epoch runs on ImageNet1k and ImageNet100 in Table 15. We see that a lower weighting term $\alpha < 0.5$ can be better for SACLR-1.

Table 9: **Computational costs on MNIST**. GPU peak memory usage from large batch training settings often used with the LARS (You et al., 2017) optimizer on MNIST with different batchsize $B$. Now we use only a linear layer with 128 units to isolate the impact of loss functions as much as possible. We downloaded the PyTorch implementations from SogCLR (Yuan et al., 2022) and iSogCLR (Qiu et al., 2023) in this study. We used a NVIDIA H100 80GB GPU and 10 core CPU with 32GB. All values over 80GB are OOM-errors.

| Method $\backslash B$ | 128 | 1K | 2K | 4K | 8K | 16K | 32K |
|---|---|---|---|---|---|---|---|
| SACLR-1 | 0.80 GB | 0.82 GB | 0.84 GB | 0.86 GB | 0.90 GB | 1.03 GB | 1.37 GB |
| SACLR-all | 0.80 GB | 0.92 GB | 1.20 GB | 2.31 GB | 6.85 GB | 24.93 GB | >80GB |
| SimCLR | 0.80 GB | 0.88 GB | 1.05 GB | 1.75 GB | 4.58 GB | 15.89 GB | 61.02 GB |
| SogCLR | 0.80 GB | 0.97 GB | 1.38 GB | 3.06 GB | 9.83 GB | 36.90 GB | >80GB |
| iSogCLR | 0.80 GB | 0.90 GB | 1.18 GB | 2.31 GB | 6.84 GB | 24.89 GB | >80GB |

Table 10: **Runtime and computational costs**. Time complexity and peak memory usage during 100 epoch training on CIFAR10 with the highest batchsize possible 16384 for a NVIDIA H100 80GB GPU without any OOM error. Here $M$ is the number of negative samples per image in batch. We used the respective PyTorch implementations from SogCLR (Yuan et al., 2022) and iSogCLR (Qiu et al., 2023) in this study. The network is a ResNet18.

| Method | Time / 100 epochs | Peak GPU memory | $M$ |
|---|---|---|---|
| SACLR-1 matrix-method (ours) | 0.44h | 27.12 GB | 1 |
| SACLR-1 row-method (ours) | 0.44h | 27.12 GB | 1 |
| SACLR-all matrix-method (ours) | 0.43h | 39.44 GB | 16384×2 - 2 |
| SACLR-all row-method (ours) | 0.44h | 39.44 GB | 16384×2 - 2 |
| SimCLR (Chen et al., 2020a) | 0.43h | 30.42 GB | 16384×2 - 2 |
| SogCLR (Yuan et al., 2022) | 0.50h | 50.92 GB | 16384×2 - 2 |
| iSogCLR (Qiu et al., 2023) | 0.48h | 38.92 GB | 16384×2 - 2 |

## A.5 Proof of Theorem 1

*Proof.* When $s_{2(i-1)+u}^{-1} = \sum_{j=1}^{N} \sum_{v=1}^{2} \mathbb{1}_{[i \neq j \text{ or } u \neq v]} q_{ij}^{u,v}, \forall i, u$

$$\mathcal{L}_{\text{CLR-row}}(\theta) = \sum_{i=1}^{N} \left[ -2 \log q_{ii}^{1,2} + \sum_{u=1}^{2} s_{2(i-1)+u} \sum_{j=1}^{N} \sum_{v=1}^{2} \mathbb{1}_{[i \neq j \text{ or } u \neq v]} q_{ij}^{u,v} \right] - 2N - \sum_{a=1}^{2N} \log s_a \tag{13}$$

$$= \left( \sum_{i=1}^{N} -2 \log q_{ii}^{1,2} \right) + 2N - 2N - \sum_{a=1}^{2N} \log s_a \tag{14}$$

$$= -\sum_{i=1}^{N} \sum_{u=1}^{2} \log q_{ii}^{1,2} + \sum_{i=1}^{N} \sum_{u=1}^{2} \log \sum_{j=1}^{N} \sum_{v=1}^{2} \mathbb{1}_{[i \neq j \text{ or } u \neq v]} q_{ij}^{u,v} \tag{15}$$

$$= -\sum_{i=1}^{N} \sum_{u=1}^{2} \log \frac{\log q_{ii}^{1,2}}{\sum_{j=1}^{N} \sum_{v=1}^{2} \mathbb{1}_{[i \neq j \text{ or } u \neq v]} q_{ij}^{u,v}} \tag{16}$$

$\square$

Table 11: **Time complexity** Time complexity of updating $s$ on CIFAR100 over 100 epochs for SACLR-1 matrix. Each run were on a RTX 4080 12GB.

|  | Normal | Constant |
|---|---|---|
| Time / 100EP | 0.67 h | 0.67 h |

Table 12: **Row-method impact of forgetting rates $\rho$ and $M$ negative samples per anchor.** Linear classification accuracy on Imagenette after 400 epochs pretraining with ResNet50 and batchsize $B = 128$. We set $\alpha = 0.5$ for each method. We report accuracy on a separate validation set split from the training set.

|  | $M = 1$ | | | $M = 2 \times 128 - 2$ | | |
|---|---|---|---|---|---|---|
| $\rho$ | 0.9 | 0.99 | 0.999 | 0.9 | 0.99 | 0.999 |
|  | **89.334** | 89.281 | 85.480 | **90.337** | 90.126 | 89.229 |

Table 13: **Matrix-method impact of forgetting rates $\rho$ and $M$ negative samples per anchor.** Linear classification accuracy on Imagenette after 400 epochs pretraining with ResNet50 and batchsize $B = 128$. We set $\alpha = 0.5$ for each method. We report accuracy on a separate validation set split from the training set.

|  | $M = 1$ | | | $M = 2 \times 128 - 2$ | | |
|---|---|---|---|---|---|---|
| $\rho$ | 0.9 | 0.99 | 0.999 | 0.9 | 0.99 | 0.999 |
|  | 89.493 | **89.915** | 89.704 | 90.443 | **90.549** | 90.285 |

## A.6 ALGORTIHMS

Table 14: **Impact of row-based estimation method compared to matrix-based estimation**. Linear classification accuracy from pretraining with batchsize $B = 128$ and $M$ negative samples per anchor, where we set by default $\alpha = 0.5$. We pretrain on ImageNet1k for 100 epochs and for 400 epochs on ImageNet100.

| | $M = 1$ | | $M = 2 \times 128 - 2$ | |
|---|---|---|---|---|
| | Row-method | Matrix-method | Row-method | Matrix-method |
| ImageNet100 (400EP) | 81.240 | 81.640 | 81.880 | 81.100 |
| ImageNet1k (100EP) | 63.638 | 63.560 | 64.053 | 63.594 |

Table 15: **Impact of weighting term** $\alpha$. Ablation studies impact of weighting term on SACLR with $M = 1$ negative samples per anchor.

| | Row-method | | Matrix-method | |
|---|---|---|---|---|
| | 0.125 | 0.5 | 0.125 | 0.5 |
| ImageNet100 (400Ep) | **81.70** | 81.24 | **82.67** | 81.64 |
| ImageNet1k (100Ep) | **64.71** | 63.63 | **64.75** | 63.56 |

---

**Algorithm 2** SACLR algorithm (using matrix-wise approximation)

---

**Input:** Input data $\{\mathbf{x}_i\}_{i=1}^N$, weighting rate $\alpha \in [0, 1]$, forgetting rate $\rho \in (0, 1)$, number of iterations $T$, batchsize $B$, number of negative samples $M$, a neural network $f$ with parameters $\theta$.

1: Initialize the neural network $f$ and $s = 1/N^2$
2: **for** $t = 1, \ldots, T$ **do**
3:      Uniformly draw $\mathcal{B} = \{i_1, \ldots, i_B\} \subset \{1, \ldots, N\}$
4:      Augment $\mathbf{x}_i$ into $\tilde{\mathbf{x}}_i^{(1)}$ and $\tilde{\mathbf{x}}_i^{(2)}$, $\forall i \in \mathcal{B}$
5:      Compute positive pairs $\{q_{ii}^{1,2} = q\big(f(\tilde{\mathbf{x}}_i^{(1)}; \theta), f(\tilde{\mathbf{x}}_i^{(2)}; \theta)\big)\}$ for each $i \in \mathcal{B}$
6:      $\mathcal{L} \leftarrow 0, \xi \leftarrow 0$
7:      **for** $i \in \mathcal{B}$ **do**
8:          Uniformly draw $\mathcal{M}_i = \{j_1, \ldots, j_M\} \subseteq \mathcal{B}$
9:          Compute negative pairs $q_{ij}^{u,v} = q\big(f(\tilde{\mathbf{x}}_i^{(u)}; \theta), f(\tilde{\mathbf{x}}_j^{(v)}; \theta)\big)$ $\forall j \in \mathcal{M}_i, u, v \in \{1, 2\}$
10:         $\mathcal{L} \leftarrow \mathcal{L} - 2\log q_{ii}^{1,2} + s \frac{N}{M} \sum_{j \in \mathcal{M}_i} \sum_{u=1}^2 \sum_{v=1}^2 q_{ij}^{u,v}$
11:         $\xi \leftarrow \xi + \frac{N^2}{B} \sum_{i \in \mathcal{B}} \left(2\alpha q_{ii}^{1,2} + (1 - \alpha) \frac{1}{M} \sum_{j \in \mathcal{M}_i} \sum_{u=1}^2 \sum_{v=1}^2 q_{ij}^{u,v}\right)$
12:      **end for**
13:      $s^{-1} \leftarrow \rho s^{-1} + (1 - \rho)\xi$
14:      Update $\theta$ with $\nabla_\theta \mathcal{L}$ using an SGD or Adam-style step.
15: **end for**
**Output:** Trained neural network $f$.

---

