# OpenReview forum: "Stochastic Approximation to Contrastive Learning"
_ICLR.cc/2025/Conference — Submitted to ICLR 2025_

### Official Review · Reviewer_gz9D · 2024-11-03

**Soundness:** 2
**Presentation:** 2
**Contribution:** 1
**Rating:** 3
**Confidence:** 5

**Summary:**

This study proposes a method, SACLR, inspired by Stochastic Cluster Embedding (SCE), a similarity-based nonlinear dimensionality reduction (NLDR) technique. SACLR reformulates contrastive learning as a matrix approximation problem using I-divergence, an unnormalized form of Kullback-Leibler divergence.

**Strengths:**

This study proposes the SACLR method, which enhances existing contrastive learning approaches. Representations learned through SACLR are optimized to support effective clustering.

**Weaknesses:**

This study lacks novelty in its methodology. SACLR appears to be a straightforward application of SCE, incorporating a few modifications to SCE. However, the study does not sufficiently explain why these modifications are important or what roles they play within the model framework.

Theorem 1, the only theoretical result presented in this study, fails to justify why SACLR would be expected to outperform SimCLR. Since Theorem 1 does not highlight any specific advantages of SACLR over existing contrastive or non-contrastive methods, including SimCLR, the theoretical contribution appears limited in its significance.

From an experimental standpoint, there is also an inadequate comparison of SACLR’s performance across diverse benchmarks. Many studies have introduced methods that significantly improve upon SimCLR, yet these methods are omitted from the benchmark comparisons in this study.

Additionally, the study lacks a comprehensive ablation analysis. Although it claims that adjusting the scaling factor can reduce computational waste, there is neither a theoretical explanation for why this adjustment reduces computation nor a rigorous ablation study to support this claim. Similarly, while the study asserts that SACLR performs well with small batch sizes, it does not provide an extensive ablation analysis to substantiate this claim.

**Questions:**

In Figure 1, this study demonstrates that SACLR achieves effective clustering. However, does strong clustering necessarily indicate a good representation? Additionally, is there evidence that SACLR performs well on various downstream tasks beyond clustering?

---

> ### Author Response · Authors · 2024-11-14
> **Which additional contrastive learning methods would you like us to include in the comparison?**
>
> Regarding your third point in the Weakness section: "From an experimental standpoint, there is also an inadequate comparison of SACLR’s performance across diverse benchmarks. Many studies have introduced methods that significantly improve upon SimCLR, yet these methods are omitted from the benchmark comparisons in this study." We have already included SimSiam, SogCLR, and iSogCLR. Could you please specify which additional contrastive learning methods should be incorporated into the empirical comparison? We are eager to hear your suggestions and will work on adding the corresponding experiments within the next two weeks.

---

> > ### Comment · Reviewer_gz9D · 2024-11-15
> >
> > Contrary to the authors' claim, SACLR has not been comprehensively benchmarked against SimSiam, SogCLR, and iSogCLR. Specifically, SimSiam results are missing from Tables 1 and 2, iSogCLR results are absent from Tables 2 and 3, and SogCLR is not included in Table 3.
> >
> > In response to the authors' question regarding additional contrastive learning methods to incorporate, it is recommended to incorporate the top-performing benchmarks from Table 1 in [A], which is already referenced in your manuscript. Notably, the results for SimCLR, MoCo v2, SimSiam, SwAV, InfoMin Aug, OBoW, BYOL, SwAV, Barlow Twins, and VICReg in Table 1 of [A] surpass those presented in Table 2 of the authors' manuscript. Additionally, referencing Tables A2 and A3 in [B] is encouraged, as they demonstrate that SimCLR can achieve higher performance than the results shown in your Table 2. For ImageNet-100 results in Table 1, it would be valuable to include results from [C].
> >
> > Finally, to substantiate claims of your method’s superiority, it is strongly recommended to conduct additional experiments in semi-supervised learning, transfer learning for image classification, and transfer learning for object detection and instance segmentation (see Tables 2, 3, and 4 in [D] for reference).
> >
> > [A] Bardes, A., Ponce, J., & LeCun, Y. (2021). Vicreg: Variance-invariance-covariance regularization for self-supervised learning. arXiv preprint arXiv:2105.04906.
> >
> > [B] Chen, T., Luo, C., & Li, L. (2021). Intriguing properties of contrastive losses. Advances in Neural Information Processing Systems, 34, 11834-11845.
> >
> > [C] Kalantidis, Y., Sariyildiz, M. B., Pion, N., Weinzaepfel, P., & Larlus, D. (2020). Hard negative mixing for contrastive learning. Advances in neural information processing systems, 33, 21798-21809.
> >
> > [D] Zbontar, J., Jing, L., Misra, I., LeCun, Y., & Deny, S. (2021, July). Barlow twins: Self-supervised learning via redundancy reduction. In International conference on machine learning (pp. 12310-12320). PMLR.

---

> > > ### Author Response · Authors · 2024-11-30
> > >
> > > We thank the reviewer for taking the time to provide feedback on our work. Below we have addressed your concerns:
> > >
> > > **Q:** This study lacks novelty in its methodology. SACLR appears to be a straightforward application of SCE, incorporating a few modifications to SCE. However, the study does not sufficiently explain why these modifications are important or what roles they play within the model framework.
> > >
> > > **A:** No, SCE cannot directly be applied to contrastive learning, as we described in the beginning of Section 3.2. Our focus is especially on solving the large batch-size challenge in contrastive learning (see abstract), which does not exist in SCE at all.
> > >
> > > **Q:** Theorem 1, the only theoretical result presented in this study, fails to justify why SACLR would be expected to outperform SimCLR. Since Theorem 1 does not highlight any specific advantages of SACLR over existing contrastive or non-contrastive methods, including SimCLR, the theoretical contribution appears limited in its significance.
> > >
> > > **A:** This is a misreading. Theorem 1 connects SACLR-row and SimCLR with a specific choice of $s$. SACLR-row is more advantageous over SimCLR because it uses a different choice of $s$ (see the remark below the theorem). Why this different choice can bring improved performance is analyzed at the end of Section 3.2.
> > >
> > >
> > >
> > >
> > > **Q:** Additionally, the study lacks a comprehensive ablation analysis. Although it claims that adjusting the scaling factor can reduce computational waste, there is neither a theoretical explanation for why this adjustment reduces computation nor a rigorous ablation study to support this claim. Similarly, while the study asserts that SACLR performs well with small batch sizes, it does not provide an extensive ablation analysis to substantiate this claim.
> > >
> > > **A:** We have added analysis and experimental results on the computational savings. See Tables 7 to 11 in Appendix.
> > > - Table 7 lists $M$ and additional variables for the compared methods.
> > > - Table 8 compares the methods for CIFAR with growing batch sizes using ResNet18.
> > > - Table 9 compares the methods for MNIST with growing batch sizes using a simple linear layer.
> > > - Table 10 records the runtime per 100 epochs and peak memory consumption for different methods and different $M$'s.
> > > - Table 11 shows that updating $s$ is very cheap.
> > >
> > >
> > >
> > > **Q:**
> > > To substantiate claims of your method’s superiority, it is strongly recommended to conduct additional experiments in semi-supervised learning, transfer learning for image classification, and transfer learning for object detection and instance segmentation (see Tables 2, 3, and 4 in [D] for reference).
> > >
> > > [A] Bardes, A., Ponce, J., & LeCun, Y. (2021). Vicreg: Variance-invariance-covariance regularization for self-supervised learning. arXiv preprint arXiv:2105.04906.
> > >
> > > [B] Chen, T., Luo, C., & Li, L. (2021). Intriguing properties of contrastive losses. Advances in Neural Information Processing Systems, 34, 11834-11845.
> > >
> > > [C] Kalantidis, Y., Sariyildiz, M. B., Pion, N., Weinzaepfel, P., & Larlus, D. (2020). Hard negative mixing for contrastive learning. Advances in neural information processing systems, 33, 21798-21809.
> > >
> > > [D] Zbontar, J., Jing, L., Misra, I., LeCun, Y., & Deny, S. (2021, July). Barlow twins: Self-supervised learning via redundancy reduction. In International conference on machine learning (pp. 12310-12320). PMLR.
> > >
> > > **A:**
> > > We have provided consistent benchmarks against SimCLR and SogCLR for all datasets. Due to time constraints in paper revision, we have completed a comparison with SimCLR, SogCLR, iSogCLR, SimSiam, and a more recent method ReSSL (suggested by Reviewer odkB) for CIFAR with both linear and the $k$NN evaluations (suggested by Reviewer Tx4K). ImageNet1k results for the additional compared methods will come later. See Tables 3 and 4, where we can see that SACLR outperforms SimSiam and ReSSL.
> > >
> > > It is important to notice that the contrastive methods from [A, B, C] used much bigger batch sizes and longer training or memory banks, synthetic data generation. For example, SimCLR from Table 1 in [A] is from the original paper and uses a batch size of 4096 and trained for 1000 epochs. In contrast, our reported results used only a batch size 128 and a few hundred epochs for ImageNet. Our goal is not to use good advantage of techniques such as  multi-crop image data augmentation strategies (Caron et., 2020), memory banks or momentum-updates to push a new state-of-the-art on image classification for each dataset. Instead, the focus is to present more efficient methods for contrastive learning. Hyperparameters are neither heavily tuned for each dataset and are kept to the settings from iSogCLR on ImageNet.

---

> ### Author Response · Authors · 2024-12-04
> **Fine-tuned results on ImageNet100**
>
> We have recently added comparison results using linear probing, kNN probing, and semi-supervised fine-tuning (as suggested in [D]) on ImageNet100. Please refer to our latest comment on Review Tx4K for further details. The results show that our method (SACLR-1) consistently outperforms SogCLR and iSogCLR.

---

### Official Review · Reviewer_C43T · 2024-11-03

**Soundness:** 3
**Presentation:** 2
**Contribution:** 2
**Rating:** 5
**Confidence:** 3

**Summary:**

This paper addresses the problem of large batch sizes required for training networks under contrastive loss. The authors use ideas from Stochastic Cluster Embedding to represent the contrastive loss as a matrix approximation problem with approximation quality measures by I-divergence, an unnormalized form of KL divergence. To adapt the Stochastic Cluster Embedding setup to representation learning under contrastive loss, the authors represent the embedding using a neural network. The authors provide two algorithms: one with a row-wise scaling/normalization factor (similar to SimCLR under uniform normalization) and one with a single scaling/normalization factor over the whole matrix. In both cases, the authors treat the scaling factor as a constant during gradient computation, with the scaling factor updated after every iteration using a weighted sum of the values of the embedding matrix. The weighted formulation of the scaling factor allows the objective to emphasize the positive samples as training progresses.

The authors also apply the proposed methods on CIFAR and Imagenet datasets to demonstrate the superior performance of the proposed methods compared to other current SOTA Contrastive Learning methods.

**Strengths:**

The paper also addresses an important problem concerning the scalability of Contrastive Learning algorithms. The paper draws inspiration from the Stochastic Cluster Embedding paper to propose a novel formulation of the constrastive learning loss function.
The proposed weighting function $w$ to calculate the scaling factor $s$ during training provides a novel and interesting way to adaptively trade off the weight of signals from the positive and negative pairs during training.

**Weaknesses:**

While the theoretical motivation of the paper is quite interesting, it is not clear why the scaling quantity $s$ is considered to be a constant in the loss function. As seen later in the paper, $s$ is a function of $q^{uv}_{ij}$ and changes during training.

While the experiments on CIFAR and Imagenet show a slight performance improvement, the experiments do not include any variance bounds on the reported metrics. Without any confidence bounds, the proposed methods' relative performance boost is unclear.

Furthermore, the paper doesn't include any experimental data on the runtime or memory improvements the proposed method provides. It is unclear how much efficiency can be gained using $M=1$ instead of $M>1$ negative samples in a batch. It would be great to see the performance differences as well as runtime differences between the baselines and the proposed method under different values of $M$.

**Questions:**

Please refer to the Weakness Section for the questions.

---

> ### Author Response · Authors · 2024-11-30
>
> We would like to thank the reviewer for your insightful comments. We have revised the research paper in the light of the suggestions and comments and hope our revision has improved the paper to a satisfactory level. We politely ask for your reconsideration. Below we have addressed your concerns:
>
>
> **Q:** While the theoretical motivation of the paper is quite interesting, it is not clear why the scaling quantity $s$ is considered to be a constant in the loss function. As seen later in the paper, $s$ is a function of $q_{ij}^{uv}$
> and changes during training.
>
> **A:** We interleave the optimization of $D_\text{I}(p||sq)$ for a fixed $s$ and periodical updates of $s$. We have revised the text to make this strategy explicit (Lines 169-170, Line 208, and Lines 295-296). In theory, one could directly substitute $s$ as a function of $q_{ij}^{uv}$ to $D_\text{I}(p||sq)$, which, however, performs worse than the interleaving strategy in practice.
>
> **Q:** While the experiments on CIFAR and Imagenet show a slight performance improvement, the experiments do not include any variance bounds on the reported metrics. Without any confidence bounds, the proposed methods' relative performance boost is unclear.
>
> **A:** Thanks. We have added the standard deviation across multiple runs of our method SACLR-1 for all datasets.
>
>
>
> **Q:** Furthermore, the paper doesn't include any experimental data on the runtime or memory improvements the proposed method provides. It is unclear how much efficiency can be gained using $M=1$ instead of $M>1$ negative samples in a batch. It would be great to see the performance differences as well as runtime differences between the baselines and the proposed method under different values of $M$.
>
> **A:** We have added comparisons on computational expenses in Tables 7-10.
> - Table 7 lists $M$ and additional variables for the compared methods.
> - Table 8 compares the methods for CIFAR with growing batch sizes using ResNet18.
> - Table 9 compares the methods for MNIST with growing batch sizes using a simple linear layer.
> - Table 10 records the runtime per 100 epochs and peak memory consumption for different methods and different $M$'s.

---

> ### Author Response · Authors · 2024-12-04
> **Fine-tuned results on ImageNet100**
>
> Please notice that we have added comparison results using linear probing, kNN probing, and semi-supervised fine-tuning (as suggested by Reviewer gz9D) on ImageNet100. Please refer to our latest comment on Review Tx4K for further details. The results show that our method (SACLR-1) consistently outperforms SogCLR and iSogCLR.

---

### Official Review · Reviewer_odkB · 2024-11-07

**Soundness:** 4
**Presentation:** 3
**Contribution:** 3
**Rating:** 6
**Confidence:** 4

**Summary:**

This paper investigates the contrastive learning paradigm, which is a typical and important representation learning method. Concretely, this paper focuses on its substantial computation resources and inspired by the Stochastic Cluster Embedding (SCE) method, authors reformulate contrastive learning as a matrix approximation problem, thus the objective function can be decomposable over instance pairs and generalize well for smaller batchsize. Extensive experiments are conducted on the CIFAR and ImageNet datasets to validate the effectiveness of the proposed loss.

**Strengths:**

+ The investigated problem is meaningful.
+ The writing is good. The derivation of the mathematical formula is convincing.
+ The experimental results are comprehensive.

**Weaknesses:**

Overall, this paper is readable and the proposed loss is convincing. But I still have some concerns as follows.

- The logic behind the proposed loss over the robustness to the fewer negative samples is unclear. Since B refers to the batchsize, I do not see the analysis over various B in the formula or theorem.

- The computational complexity should be analyzed or empirically investigated since it needs to calculate s in Eq.(6).

- The baseline methods are sort of weak. Since it focuses on the smaller batchsize, some other competing methods in terms of smaller batchsize should be included as well. For example, ReSSL (NeurIPS 2021) achieves 69.9% (Table 6) on ImageNet with ResNet50 which is much higher than that in this paper.

**Questions:**

Please refer to the Weaknesses.

---

> ### Author Response · Authors · 2024-11-30
>
> We would like to thank the reviewer for your insightful comments. Below we have addressed your concerns:
>
>
> **Q:** The logic behind the proposed loss over the robustness to the fewer negative samples is unclear. Since B refers to the batchsize, I do not see the analysis over various B in the formula or theorem.
>
> **A:** We have added Figure 1 to clarify the development process. The key insight is that negative pairs provide only little learning signals. Reducing the number of negative pairs can significantly lower computational costs, as demonstrated in Tables 7 to 10. However, simple downsampling increases the variance of gradient approximations. To address this, we adopt a more flexible divergence to handle $q$ and $s$ separately. We then introduce a generalized $s$ with adaptive, non-uniform weights to minimize the impact of negative pairs (see Section 3.2, last paragraph). Finally, we apply stochastic approximation for robust mini-batch optimization.
>
> **Q:** The computational complexity should be analyzed or empirically investigated since it needs to calculate s in Eq.(6).
>
> **A:** Computing $s$ is cheap. We have added a comparison in Appendix Table 11, where updating $s$ or keeping it constant has a negligible time difference.
>
>
> **Q:** The baseline methods are sort of weak. Since it focuses on the smaller batchsize, some other competing methods in terms of smaller batchsize should be included as well. For example, ReSSL (NeurIPS 2021) achieves 69.9\% (Table 6) on ImageNet with ResNet50 which is much higher than that in this paper.
>
> **A:** We have incorporated comparisons with ReSSL in Tables 3 and 4.
>
>
> It is important to note that our primary goal is not to push the state-of-the-art ImageNet classification accuracy. We did not extensively fine-tune the SACLR hyperparameters for each dataset. Instead, we adopted the hyperparameter settings from iSogCLR and applied the same configurations consistently across all tested datasets.

---

### Official Review · Reviewer_Tx4K · 2024-11-08

**Soundness:** 2
**Presentation:** 3
**Contribution:** 3
**Rating:** 5
**Confidence:** 3

**Summary:**

This work proposes an approach to reformulate contrastive learning, used in many self-supervised learning methods such as SimCLR, in terms of matrix approximations. This reformation enables stochastic approximations of contrastive learning to enable good performance with smaller batch sizes. The authors validate the idea with experiments on CIFAR and ImageNet by measuring linear evaluation classification accuracy as well as embedding projections. The authors compare the method against standard self-supervised training method such as SimCLR as well as stochastic approximation methods such as SogCLR that also aim to reduce the need for large batch sizes in self-supervised training.

**Strengths:**

The authors study the important problem of revisting contrastive learning's reliance on large batch sizes. The authors present an interesting reformulation of contrastive learning by taking inspiration from stochastic cluster embeddings. This point of view is an interesting perspective on standard contrastive learning
Experimentally, the authors train SACLR on CIFAR and ImageNet using standard setup and compare against a reasonable set of baselines. I appreciate the author's notes on which libraries were used for the method implementations, aiding reproducibility (although I do hope the authors also release code for the proposed method).

**Weaknesses:**

# Presentation
* The introduction is quite lengthy and rehashes broadly well-known statements about the role of deep learning. I'd recommend the authors trim this lengthy introduction.
* With this gained space, I recommend 1) the authors describe in more detail the similarities and differences to existing stochastic methods such as SogCLR that also tackle the same challenge of large batch sizes in contrastive learning. 2) the authors consider illustrating the method visually with a method figure. Right now the method as presented in Section 3 and Algorithm 1 requires parsing many variables, with nested summations that muddy the overall idea and intuition. A high-level intuitive figure to illustrate the method can help a great deal in this regard.
* The performance gains on ImageNet are marginal compared to SogCLR (and I'd be curious how they compare to iSogCLR as well). While this doesn't diminish the contribution in itself, I do recommend the authors downtone the claims made in the introduction and abstract accordingly.

# Claims
* The authors claim our method is "more computationally efficient" (line 72), "signficant computaitonal resources are wasted on negative pairs that offer little learnign signal" (line 61), and again in the abstract (line 24). This claim is missing exactly what the efficiency is relative to—is it existing efficient methods such as SogCLR or standard SimCLR? This needs clarification and evidence. To support this claim the authors should provide evidence of the GPU training hours, flops, or amount of memory saved, which right now I could not find.

* The authors state SCE requires fixed input similarites and cannot generalize to data points outside the training set (lines 177-179), after which they suggest their method overcomes these limitations with three modification (using a deep neural network, using augmented view and row-ise vector approximation), however, no further discussion of how these three modification address the two limitations highlighted is provided. Can you the authors respond provide a detailed justification for this claim?

- The authors only provide classification accuracy figures using linear evaluation. I recommend the authors also include other evaluation common approaches such as finetuning, KNN-evaluation, and retrieval (see for example SogCLR) to provide a more complete picture of the methods' benefits.

**Questions:**

- Performance on ImageNet100, I'm curious why the authors suspect the matrix versus row method difference is much larger on ImageNet100 compared to ImageNet?
- I'm confused by what the comparison in Figure 1 shows. Is t-SNE panel illustrating standard SimCLR feature projections compared to SACLR? If so, maybe a better label is warranted?
- Which dataset is used in the ablations for figure 2? I'm also curious how the linear evaluation accuracy compares for SACLR-1 versus SACLR-all.

---

> ### Author Response · Authors · 2024-11-30
>
> We would like to thank the reviewer for your insightful comments. We have revised the paper in the light of the suggestions and comments and hope our revision has improved the paper to a satisfactory level. We politely ask for your reconsideration. Below we have addressed your concerns:
>
> **Q:** The introduction is quite lengthy and rehashes broadly well-known statements about the role of deep learning. I'd recommend the authors trim this lengthy introduction. With this gained space, I recommend 1) the authors describe in more detail the similarities and differences to existing stochastic methods such as SogCLR that also tackle the same challenge of large batch sizes in contrastive learning. 2) the authors consider illustrating the method visually with a method figure. Right now the method as presented in Section 3 and Algorithm 1 requires parsing many variables, with nested summations that muddy the overall idea and intuition. A high-level intuitive figure to illustrate the method can help a great deal in this regard.
>
> **A:** Thanks for the advice. The introduction has been trimmed, with SogCLR added. We have also added Figure 1 to illustrate the development clue, giving a big picture for the readers.
>
>
> **Q:** The performance gains on ImageNet are marginal compared to SogCLR (and I'd be curious how they compare to iSogCLR as well). While this doesn't diminish the contribution in itself, I do recommend the authors downtone the claims made in the introduction and abstract accordingly.
>
> **A:** We have downtoned the claims about ImageNet in Sections 1, 4, and 6.
>
> **Q:** The authors claim our method is "more computationally efficient" (line 72), "signficant computaitonal resources are wasted on negative pairs that offer little learnign signal" (line 61), and again in the abstract (line 24). This claim is missing exactly what the efficiency is relative to—is it existing efficient methods such as SogCLR or standard SimCLR? This needs clarification and evidence. To support this claim the authors should provide evidence of the GPU training hours, flops, or amount of memory saved, which right now I could not find.
>
> **A:** SACLR has significant advantages over SimCLR and SogCLR in memory consumption. We have added evidence to support our claim (see Tables 8, 9, and 10).
>
>
>
> **Q:** The authors state SCE requires fixed input similarites and cannot generalize to data points outside the training set (lines 177-179), after which they suggest their method overcomes these limitations with three modification (using a deep neural network, using augmented view and row-ise vector approximation), however, no further discussion of how these three modification address the two limitations highlighted is provided. Can you the authors respond provide a detailed justification for this claim?
>
> **A:** We have revised the part (beginning of Section 3.2), making the correspondence between limitations and modifications more explicit.
>
>
> **Q:** The authors only provide classification accuracy figures using linear evaluation. I recommend the authors also include other evaluation common approaches such as finetuning, KNN-evaluation, and retrieval (see for example SogCLR) to provide a more complete picture of the methods' benefits.
>
> **A:** We have added the $k$NN evaluation in Table 4, where SACLR still wins for CIFAR, and Imagenette.
>
>
> **Q:** Performance on ImageNet100, I'm curious why the authors suspect the matrix versus row method difference is much larger on ImageNet100 compared to ImageNet?
>
> **A:** The difference is expected because we did not tune the hyperparameters with substantial effort. We simply used the same hyperparameters for SACLR on all datasets.
>
>
>
>
> **Q:** I'm confused by what the comparison in Figure 1 shows. Is t-SNE panel illustrating standard SimCLR feature projections compared to SACLR? If so, maybe a better label is warranted?
>
> **A:**
> We have clarified the visualizations. The new Figure 2 shows embedding visualizations of Imagenette dataset where all compared methods used the Cauchy-kernel.
>
> **Q:** Which dataset is used in the ablations for figure 2? I'm also curious how the linear evaluation accuracy compares for SACLR-1 versus SACLR-all.
>
> **A:**
> The visualizations use the Imagenette dataset, a subset of ImageNet with 10 different classes. More comparison results for CIFAR can be found in Tables 3 and 4, while more results for ImageNet are given in Tables 5 and 6 in Appendix.

---

> ### Author Response · Authors · 2024-12-03
>
> We did not have enough time to complete additional results on ImageNet100 before now. We include here additional results from our reproductions on semi-supervised finetuning following [D] (suggested from Reviewer gz9D) and kNN on ImageNet100. Results will be included to paper. We find consistent improvements over SogCLR over kNN , linear and semi-supervised evaluations on ImageNet100. For semi-supervised evaluations we used subsets of size 1\% and 10\% using the split and source code from [D] on ImageNet100.
>
>
> |	      |   Top 1   |           |	               |	           |		Top 5   |               |
> |---------|-----------|-----------|----------------|---------------|----------------|---------------|
> | Method  | Linear    |    kNN    | fine-tune 1\%  | fine-tune 10\%| fine-tune 1\%  | fine-tune 10\%|
> |SACLR-1  |   82.64   |  77.30    |   67.50        | 78.720        |   90.280       |  95.08        |
> |SogCLR   |   80.72   |  74.90    |   63.98        | 77.16         |   87.600       |  93.76        |
> |iSogCLR  |   81.72   |  75.62    |   68.50        | 76.76         |   89.50        |  93.78        |
>
>
>
> [D] Zbontar, J., Jing, L., Misra, I., LeCun, Y., & Deny, S. (2021, July). Barlow twins: Self-supervised learning via redundancy reduction. In International conference on machine learning (pp. 12310-12320). PMLR.

---

### Meta-Review · Area_Chair_9uhA · 2024-12-20

**Metareview:**

This paper reformulates contrastive learning as a matrix approximation problem using I-divergence. The purpose is to enable good performance with smaller batch sizes. Experiments are performed on CIFAR and ImageNet to validate the idea. This approach is compared to SogCLR, ReSSL, and other baselines.

Initial concerns raised by reviewers include: the baselines are weak and the improvement is marginal; the claim of "more efficient" is vague; some analysis is missing, e.g., the computational complexity; there is an inadequate comparison of SACLR’s performance across diverse benchmarks -- many studies that significantly improve upon SimCLR, are omitted.

**Additional Comments On Reviewer Discussion:**

During rebuttal, some questions were to be answered by the authors, such as the memory usage of the approach. More baselines have been added.

Overall, the comparison is still inadequate, e.g., the comparison on ImageNet only includes very weak baselines, and it is hard to see how the improvement in this proposed paper fits into more recent studies.

---

### Decision · Program_Chairs · 2025-01-22

Reject